# NEARLY-OPTIMAL BANDIT LEARNING IN STACKELBERG GAMES WITH SIDE INFORMATION

**Maria-Florina Balcan**[*]
Carnegie Mellon University

**Martino Bernasconi**[*]
Bocconi University

**Matteo Castiglioni**[*]
Politecnico di Milano

**Andrea Celli**[*]
Bocconi University

**Keegan Harris**[*]
University of California, Berkeley

**Zhiwei Steven Wu**[*]
Carnegie Mellon University

## ABSTRACT

We study the problem of online learning in Stackelberg games with side information between a *leader* and a sequence of *followers*. In every round the leader observes *contextual information* and *commits* to a mixed strategy, after which the follower *best-responds*. We provide learning algorithms for the leader which achieve $\tilde{O}(T^{1/2})$ regret under *bandit feedback*, an improvement from the previously best-known rates of $\tilde{O}(T^{2/3})$. Our algorithms rely on a reduction to linear contextual bandits in the utility space: In each round, a linear contextual bandit algorithm recommends a utility vector, which our algorithm inverts to determine the leader's mixed strategy. We extend our algorithms to the setting in which the leader's utility function is unknown, and also apply it to the problems of bidding in second-price auctions with side information and online Bayesian persuasion with public and private states. Finally, we observe that our algorithms empirically outperform previous results on numerical simulations.

## 1 INTRODUCTION

Many real-world strategic settings take the form of *Stackelberg games*, in which the leader *commits* to a (randomized) strategy and the follower(s) *best-respond*. For example, in security domains (e.g. airport security, wildlife protection) the leader (federal officers with drug-sniffing dogs, park rangers) chooses a patrol strategy, which the follower (drug smuggler, poacher) observes before choosing an area to exploit. In such settings, the leader may face different *follower types* over time, each with their own goals and objectives.

We study a generalization of the traditional Stackelberg game setting in which the payoffs of the players depend on additional *contextual information* (or *side information*) that is not captured in the players' actions and may vary over time. Such contextual information naturally arises in many Stackelberg game settings: In airport security, different parts of the airport may be more crowded during different parts of the day, which may make it easier or harder to smuggle items through security in those areas. In wildlife protection, different animal species may be easier or harder to poach at different times of the year, due to factors such as migration patterns and weather. More generally, side information may appear in the form of (an embedding of) natural language, such as threat reports, incident descriptions, or intelligence briefings, which can alter the leader's optimal commitment and/or the follower's strategic response.

Harris et al. (2024) formalize this setting and provide online learning algorithms for the leader when the followers and contextual information change over time. Their algorithms obtain $\tilde{O}(T^{1/2})$ regret[1] under *full feedback* (i.e. when information about the follower is revealed to the leader after each round) where $T$ is the number of time-steps, but only $\tilde{O}(T^{2/3})$ regret under the more challenging (and more realistic) bandit feedback setting, where only the follower's action is revealed.

---

[*]Authors are listed in alphabetical order. Corresponding author: `keegan.harris@berkeley.edu`.
[1]Regret is the cumulative difference between the best possible policy's utility and the leader's utility.

**Our contributions** We close the gap from $\tilde{O}(T^{2/3})$ to $\tilde{O}(T^{1/2})$ regret under bandit feedback, which matches known lower bounds up to logarithmic factors. As in Harris et al. (2024), we study two settings: one in which the sequence of contextual information is chosen adversarially and the sequence of followers is chosen stochastically, and the setting where the contextual information is chosen stochastically and the followers are chosen adversarially. Moreover, the algorithms of Harris et al. (2024) are not applicable when the follower's utility depends on the contextual information, an assumption which we do not need.

In both settings (adversarial contextual information and adversarial follower types), our algorithm (Algorithm 1) is a reduction to linear contextual bandits. While the leader's utility is a non-linear function of their strategy, we can linearize the problem by playing in the leader's "utility space". In each round, a linear contextual bandit algorithm plays a vector in the image of the leader's utility, where the $i$-th component of the vector is the leader's expected utility when facing the $i$-th follower type. The leader then plays the strategy which induces this utility vector and gives their observed reward as feedback to the contextual bandit algorithm. By reformulating the problem in this way, we can take advantage of the rich literature on linear contextual bandits. Indeed, by instantiating Algorithm 1 with different contextual bandit algorithms, we obtain regret guarantees for both settings.

Next we study an extension to the setting where the leader's utility function is unknown and must be learned over time. We show that a similar reduction to contextual bandits holds in this setting under a linearity assumption on the leader's utility function. This reduction still obtains $\tilde{O}(T^{1/2})$ regret, albeit at the cost of additional polynomial factors in the size of the problem instance in the regret bound.

In Section 4 we show how to apply our algorithm to learning in other settings which exhibit the same type of structure; specifically (i) learning in second-price auctions with side information and (ii) online Bayesian persuasion with side information. We are the first to study either of these settings, to the best of our knowledge, despite the fact that side information naturally arises in both auctions and Bayesian persuasion settings. Our results largely carry over to these applications as-is, although we need to discretize the learner's action space in a different way than we do for Stackelberg games.

Our work is conceptually related to Bernasconi et al. (2023), who use a similar reduction to obtain $\tilde{O}(T^{1/2})$ regret in online Bayesian persuasion, learning in auctions, and learning in Stackelberg games *without* side information. While their main result is a reduction to adversarial linear bandits, our problem reduces to a linear contextual bandit problem with an infinite action set. Furthermore, either the action set or the sequence of contexts will be chosen adversarially, depending on which setting we are in. While such contextual bandit problems are generally intractable, we leverage the special structure present in our setting to obtain positive results. In particular, we observe that the optimal strategy at each time-step will always belong to a time-varying, but finite, set. Therefore, we can discretize the utility space in a way that allows us to apply our reduction with two specific linear contextual bandit algorithms that allow for both time-varying action sets and adversarially-chosen contexts/action sets. Additionally, while both our reduction and theirs leverage the linear structure which is induced from having finitely-many follower types, our extension to unknown leader utilities in Section 3.3 uses a more general version of this linear structure in order to compensate for the additional uncertainty from unknown utilities. Our reduction is also more streamlined, as it does not require a per-round application of Caratheodory's Theorem, due to our discretization step.

## 1.1 RELATED WORK

**Learning in Stackelberg games** Conitzer & Sandholm (2006) provide algorithms and prove NP-Hardness results for the problem of computing equilibrium in various Stackelberg game settings when all parameters of the problem are known. A line of work on *learning* Stackelberg games Letchford et al. (2009); Peng et al. (2019); Bacchiocchi et al. (2025) relaxes the assumption that all parameters of the problem are known to the leader, and instead posits that they are given a number of (player actions, outcome) tuples to learn from.

Our work falls under the category of *online* learning in Stackelberg games, where the sequence of data arrives sequentially instead of all at once. This setting was first introduced by Balcan et al. (2015) and was generalized to handle settings with side information in Harris et al. (2024). Balcan et al. (2026) also study side information, but focus on a more structured setting where the side information is predictive of the follower's type.

Other recent work on learning in Stackelberg games includes meta-learning Harris et al. (2023a), learning in cooperative Stackelberg games (e.g. Zhao et al. (2023); Donahue et al. (2024)), strategizing against a follower who plays a no-regret learning algorithm (e.g. Braverman et al. (2018); Deng et al. (2019)), and learning various structured Stackelberg games such as strategic classification (e.g. Hardt et al. (2016); Dong et al. (2018); Harris et al. (2023b)), performative prediction (e.g. Perdomo et al. (2020); Hardt & Mendler-Dünner (2023)), and principal-agent problems (e.g. Ho et al. (2016)).

Online learning in Stackelberg games is conceptually related to the online optimization of piecewise Lipschitz functions, as the underlying reward function after fixing any fixed follower type (and piece of side information) is piecewise linear Dick et al. (2020); Sharma et al. (2020); Balcan et al. (2018). However the techniques used in this line of work are not applicable to our setting, since the follower type and contextual information change from round-to-round.

**Learning in auctions and persuasion**  Our algorithms are also applicable to generalizations of the problems of online learning in simultaneous second-price auctions Daskalakis & Syrgkanis (2022) and online Bayesian persuasion Castiglioni et al. (2020). Flajolet & Jaillet (2017) also study online learning in second-price auctions with side information. They consider single-item auction settings with a budget constraint, while we consider combinatorial auctions with no budget constraints. We also consider a more general form of adversarial feedback than they consider.

**Contextual bandits**  Finally, one may view our setting as a special type of contextual bandit problem with continuous action spaces and non-linear rewards. While one could, in principle, attempt to apply a black-box contextual bandit algorithm to our setting (e.g. Syrgkanis et al. (2016a;b); Rakhlin & Sridharan (2016)), we are not aware of any algorithms which obtain meaningful performance guarantees under this reward structure without (1) making additional assumptions about the learner's knowledge of the sequence of contexts they will face *and* (2) obtaining generally worse rates.

## 2  PRELIMINARIES

We use $\Delta(\mathcal{A})$ to denote the probability simplex over the (finite) set $\mathcal{A}$, and $[N] := \{1, \ldots, N\}$ to denote the set of integers from 1 to $N \in \mathbb{N}_{>0}$.

We study a repeated interaction between a leader and a sequence of followers over $T$ rounds. In round $t \in [T]$, both players observe a context $\mathbf{z}_t \in \mathcal{Z} \subseteq \mathbb{R}^d$, which represents the side information available (e.g. information about weather patterns, airport congestion levels) in the current round. The leader then commits to a mixed strategy $\mathbf{x}_t \in \Delta(\mathcal{A}_l)$, where $\mathcal{A}_l$ is the leader's action set and $A_l := |\mathcal{A}_l| < \infty$. After observing the context $\mathbf{z}_t$ and the leader's mixed strategy $\mathbf{x}_t$, follower $f_t$ plays action $a_{f,t} \in \mathcal{A}_f$, where $\mathcal{A}_f$ is the follower's action set and $A_f := |\mathcal{A}_f| < \infty$. The leader's action $a_{l,t}$ is then sampled according to their mixed strategy $\mathbf{x}_t$.

After the round is over, the leader receives utility $u(\mathbf{z}_t, a_{l,t}, a_{f,t})$, according to their utility function $u : \mathcal{Z} \times \mathcal{A}_l \times \mathcal{A}_f \to [-1, 1]$. Similarly, follower $f_t$ receives utility $u_{f_t}(\mathbf{z}_t, a_{l,t}, a_{f,t})$ according to utility function $u_{f_t} : \mathcal{Z} \times \mathcal{A}_l \times \mathcal{A}_f \to [-1, 1]$. We often use the shorthand $u(\mathbf{z}_t, \mathbf{x}_t, a_{f,t}) := \mathbb{E}_{a_{l,t} \sim \mathbf{x}_t}[u(\mathbf{z}_t, a_{l,t}, a_{f,t})]$ (resp. $u_{f_t}(\mathbf{z}_t, \mathbf{x}_t, a_{f,t}) := \mathbb{E}_{a_{l,t} \sim \mathbf{x}_t}[u_{f_t}(\mathbf{z}_t, a_{l,t}, a_{f,t})]$) to denote the leader's (resp. follower's) expected utility with respect to the randomness in the leader's mixed strategy.

We assume that the follower in each round is one of $K < \infty$ types $f_t \in [K]$, where follower type $i \in [K]$ corresponds to utility function $u_i$. We assume that $u_1, \ldots, u_K$ are known to the leader, but the identity of follower $f_t$ is *never* revealed.[2] This setting is referred to as *bandit feedback* in the literature on online learning in Stackelberg games Balcan et al. (2015); Harris et al. (2024).

Given a context $\mathbf{z}_t$ and leader mixed strategy $\mathbf{x}_t$, follower $f_t$'s *best-response* is $b_{f_t}(\mathbf{z}_t, \mathbf{x}_t) := \arg\max_{a_f \in \mathcal{A}_f} u_{f_t}(\mathbf{z}_t, \mathbf{x}_t, a_f)$, where ties are broken in an unknown-but-fixed way. We measure the performance of the leader via the notion of *regret*:

---

[2]Balcan et al. (2015) show that learning is impossible when $K = \infty$ in (non-contextual) Stackelberg games, which implies an impossibility result for our setting.

---

**Algorithm 1:** Reduction to Linear Contextual Bandits

---

**Input:** Linear contextual bandit algorithm $\mathcal{R}$
**for** $t = 1, \ldots, T$ **do**
    Observe $\mathbf{z}_t$, compute $U_t := \{\mathbf{u}(\mathbf{z}_t, \mathbf{x}) : \mathbf{x} \in \mathcal{E}_t\}$
    Let $\mathbf{v}_t \leftarrow \mathcal{R}.\text{recommend}(U_t)$
    Commit to the mixed strategy $\mathbf{x}_t$ which induces $\mathbf{v}_t$
    Play action $a_{l,t} \sim \mathbf{x}_t$ and call $\mathcal{R}.\text{observeUtility}(\mathbf{v}_t, u(\mathbf{z}_t, a_{l,t}, b_{f_t}(\mathbf{z}_t, \mathbf{x}_t)))$
**end**

---

**Definition 2.1** (Contextual Stackelberg Regret). *The leader's contextual Stackelberg regret with respect to context sequence $\mathbf{z}_1, \ldots, \mathbf{z}_T$ and follower sequence $f_1, \ldots, f_T$ is $R(T) := \sum_{t=1}^{T} u(\mathbf{z}_t, \mathbf{x}_t^*, b_{f_t}(\mathbf{z}_t, \mathbf{x}_t^*)) - u(\mathbf{z}_t, \mathbf{x}_t, b_{f_t}(\mathbf{z}_t, \mathbf{x}_t))$, where $\mathbf{x}_t^* = \pi^*(\mathbf{z}_t) := \arg\max_{\mathbf{x} \in \Delta(\mathcal{A}_l)} \sum_{\tau : \mathbf{z}_\tau = \mathbf{z}_t} u(\mathbf{z}_\tau, \mathbf{x}, b_{f_\tau}(\mathbf{z}_\tau, \mathbf{x}))$ is the mixed strategy played by the optimal-in-hindsight policy $\pi^*$ at time $t$.*

Since previous work Harris et al. (2024) shows that no-regret learning is impossible (i.e. there exists no algorithm for which $R(T) = o(T)$) when the sequence of contexts and the sequence of followers are chosen jointly by an adversary with knowledge of the leader's algorithm, we focus on two natural relaxations: the setting where the sequence of contexts is chosen by an adversary and the sequence of follower types are drawn from an unknown (stationary) distribution (Section 3.1) and the setting where the sequence of follower types are chosen by an adversary and the sequence of contexts are drawn from an unknown distribution (Section 3.2). All of our results are applicable to the simpler setting where both the contexts and follower types are chosen stochastically.

While the leader's action space $\Delta(\mathcal{A}_l)$ is infinitely large, we follow the lead of previous work and consider a discretization which is nearly wothout loss of generality. The following two definitions are from Harris et al. (2024).

**Definition 2.2** (Contextual Follower Best-Response Region). *For follower type $i \in [K]$, follower action $a_f \in \mathcal{A}_f$, and context $\mathbf{z} \in \mathcal{Z}$, let $\mathcal{X}_{\mathbf{z}}(i, a_f) \subseteq \Delta(\mathcal{A}_l)$ denote the set of all leader mixed strategies such that a follower of type $i$ best-responds to all $\mathbf{x} \in \mathcal{X}_{\mathbf{z}}(i, a_f)$ by playing action $a_f$ under context $\mathbf{z}$, i.e., $\mathcal{X}_{\mathbf{z}}(i, a_f) = \{\mathbf{x} \in \mathcal{X} : b_i(\mathbf{z}, \mathbf{x}) = a_f\}$.*

**Definition 2.3** (Contextual Best-Response Region). *For a given function $\sigma : [K] \to \mathcal{A}_f$, let $\mathcal{X}_{\mathbf{z}}(\sigma)$ denote the set of all leader mixed strategies such that under context $\mathbf{z}$, a follower of type $i$ plays action $\sigma(i)$ for all $i \in [K]$, i.e. $\mathcal{X}_{\mathbf{z}}(\sigma) = \cap_{i \in [K]} \mathcal{X}_{\mathbf{z}}(i, \sigma(i))$.*

It is straightforward to show that all contextual best-response regions are convex and bounded (but not necessarily closed). Because of this, the loss in performance is negligible from restricting the leader's strategy space to be the set of approximate extreme points of all contextual best-response regions. Formally, we define $\mathcal{E}_t$ as follows.

**Definition 2.4** ($\delta$-approximate extreme points). *Fix a context $\mathbf{z} \in \mathcal{Z}$ and consider the set of all non-empty contextual best-response regions. For $\delta > 0$, $\mathcal{E}_{\mathbf{z}}(\delta)$ is the set of leader mixed strategies such that for all best-response functions $\sigma$ and any $\mathbf{x} \in \Delta(\mathcal{A}_l)$ that is an extreme point of $cl(\mathcal{X}_{\mathbf{z}}(\sigma))$, $\mathbf{x} \in \mathcal{E}_{\mathbf{z}}(\delta)$ if $\mathbf{x} \in \mathcal{X}_{\mathbf{z}}(\sigma)$. Otherwise there is some $\mathbf{x}' \in \mathcal{E}_{\mathbf{z}}(\delta)$ such that $\mathbf{x}' \in \mathcal{X}_{\mathbf{z}}(\sigma)$ and $\|\mathbf{x}' - \mathbf{x}\|_1 \leq \delta$. With a slight abuse of notation, we define the set of approximate extreme points $\mathcal{E}_t$ to be $\mathcal{E}_t := \mathcal{E}_{\mathbf{z}_t}(\frac{1}{T})$.*

Balcan et al. (2015) show that $|\mathcal{E}_t| = O((K A_f^2)^{A_l} A_f^K)$. By Lemma 4.4 in Harris et al. (2024), restricting the learner to policies which only play strategies in $\mathcal{E}_t$ at round $t$ leads to at most $O(1)$ additional regret. We will use fact throughout the sequel.

## 3   A REDUCTION TO LINEAR CONTEXTUAL BANDITS

Our main result is an algorithm (Algorithm 1) that achieves $\tilde{O}(T^{1/2})$ regret in both the setting where contexts are chosen adversarially and follower types are chosen stochastically (Section 3.1) and the setting where the contexts are chosen stochastically and follower types are chosen adversarially (Section 3.2). While the leader's utility is a non-linear function of their mixed strategy $\mathbf{x}_t$ in any

given round (due to the follower's best-response $b(\mathbf{z}_t, \mathbf{x}_t)$), we can "linearize" the problem by leveraging the fact that the leader's utility can be written as $u(\mathbf{z}_t, \mathbf{x}_t, b_{f_t}(\mathbf{z}_t, \mathbf{x}_t)) = \langle \mathbf{u}(\mathbf{z}_t, \mathbf{x}_t), \mathbf{1}_{f_t} \rangle$, where $\mathbf{u}(\mathbf{z}, \mathbf{x}) := [u(\mathbf{z}, \mathbf{x}, b_1(\mathbf{z}, \mathbf{x})), \ldots, u(\mathbf{z}, \mathbf{x}, b_K(\mathbf{z}, \mathbf{x}))]^\top \in \mathbb{R}^K$ is the vector of utilities the leader would receive against each follower type given context $\mathbf{z}$ and mixed strategy $\mathbf{x}$, and $\mathbf{1}_{f_t} \in \mathbb{R}^K$ is a one-hot vector with a 1 in the $f_t$-th component and zeros elsewhere. Since $u(\mathbf{z}_t, \mathbf{x}_t, b_f(\mathbf{z}_t, \mathbf{x}_t))$ is a linear function of $\mathbf{u}(\mathbf{z}_t, \mathbf{x}_t)$, one can use an off-the-shelf linear contextual bandit algorithm to pick a vector $\mathbf{v}_t$ in the image of $\mathbf{u}(\mathbf{z}_t, \cdot)$, then invert the mapping to find the mixed strategy $\mathbf{x}_t$ such that $\mathbf{v}_t = \mathbf{u}(\mathbf{z}_t, \mathbf{x}_t)$.

Algorithm 1 takes as input a linear contextual bandit algorithm $\mathcal{R}$, which, (1) when given a (finite) set of actions $U_t$, returns an element $\mathbf{v}_t \in U_t$ ($\mathcal{R}.\text{recommend}()$) and (2) updates its internal parameters when given an action $\mathbf{v}_t$ and a realized utility $u_t \in [-1, 1]$ ($\mathcal{R}.\text{observeUtility}()$). Finally, while the leader's action space $\Delta(\mathcal{A}_l)$ is infinitely large (and thus, so is the dual space $\tilde{U}_t := \{\mathbf{u}(\mathbf{z}_t, \mathbf{x}) : \mathbf{x} \in \Delta(\mathcal{A}_l)\}$), the leader incurs essentially no loss in utility by restricting themselves to a finite (but exponentially-large) set of context-dependent points $\mathcal{E}_t$, which roughly correspond to the set of extreme points of convex polytopes which are induced by the followers' best-responses. As such, our algorithm operates on the set of utility vectors $U_t := \{\mathbf{u}(\mathbf{z}_t, \mathbf{x}) : \mathbf{x} \in \mathcal{E}_t\}$ in each round.[3]

### 3.1 ADVERSARIAL CONTEXTS AND STOCHASTIC FOLLOWER TYPES

To get no-regret guarantees when the sequence of contexts is chosen adversarially and the sequence of follower types is chosen stochastically, we instantiate Algorithm 1 with the Optimism in the Face of Uncertainty for Linear models (OFUL) linear contextual bandit algorithm of Abbasi-Yadkori et al. (2011). OFUL leverages the principle of optimism under uncertainty to balance exploration and exploitation. Specifically, it assumes a linear relationship between utilities and actions such that $\mathbb{E}[u_t] = \langle \mathbf{v}_t, \boldsymbol{\theta}^* \rangle$, where $\mathbf{v}_t \in \mathbb{R}^K$ is an action from some exogenously-given set $U_t$, and $\boldsymbol{\theta}^* \in \mathbb{R}^K$ is an unknown parameter. OFUL maintains a confidence set $C_t$ over $\boldsymbol{\theta}^*$ in round $t$ such that $\boldsymbol{\theta}^* \in C_t$ with high probability, which it updates based on the noisy observed utility $u_t$. In each round, it then selects the action that maximizes the upper confidence bound on the expected reward, i.e. it plays action $\mathbf{v}_t \in \arg\max_{\mathbf{v} \in U_t, \boldsymbol{\theta} \in C_t} \langle \mathbf{v}, \boldsymbol{\theta} \rangle$.

We show that when follower types are chosen stochastically, the leader's utility at time $t$ can be written as $u(\mathbf{z}_t, \mathbf{x}_t, b_{f_t}(\mathbf{z}_t, \mathbf{x}_t)) = \langle \mathbf{u}(\mathbf{z}_t, \mathbf{x}_t), \mathbf{p}^* \rangle + \epsilon_t$, where $\mathbf{p}^* \in \Delta^K$ is the true (unknown) distribution over follower types, and $\epsilon_t \in [-4, 4]$ is a zero-mean random variable. Therefore by instantiating Algorithm 1 with OFUL, we can optimistically learn $\mathbf{p}^*$ and attain $\tilde{O}(\sqrt{T})$ regret in this setting.

**Theorem 3.1.** *When $\mathcal{R}$ is instantiated as the OFUL algorithm of Abbasi-Yadkori et al. (2011), Algorithm 1 obtains expected contextual Stackelberg regret $\mathbb{E}[R(T)] = O(K\sqrt{T}\log(T))$ when the sequence of contexts is chosen adversarially and the sequence of follower types is chosen stochastically. The expectation is taken with respect to both the randomness in Algorithm 1, as well as the distribution over follower types.*

### 3.2 STOCHASTIC CONTEXTS AND ADVERSARIAL FOLLOWER TYPES

When the sequence of follower types is chosen adversarially, there will be no underlying distribution $\mathbf{p}^*$ over follower types for the algorithm to learn. As such, instantiating $\mathcal{R}$ with OFUL will not provide meaningful regret guarantees in this setting. Instead, we instantiate $\mathcal{R}$ using a modified version of Algorithm 1 in Liu et al. (2023) (Algorithm 2).

Algorithm 1 in Liu et al. (2023) (henceforth referred to as logdet-FTRL) uses a variant of Follow-The-Regularized-Leader with the log-determinant barrier as the regularizer to solve a variant of the linear contextual bandit problem with adversarial losses. In their setting, the learner receives a set of actions $U'_t$ in each round which are drawn from some distribution over the unit ball and plays an action $\mathbf{v}'_t \in U'_t$.[4] The learner then receives loss $\ell_t$ such that $\mathbb{E}[\ell_t] = \langle \mathbf{v}'_t, \mathbf{y}_t \rangle$, where $\mathbf{y}_t$ is chosen adversarially.

---

[3]This is important, as the regret minimizers we instantiate Algorithm 1 with in Section 3.1 and Section 3.2 both require the action set to be finite.

[4]logdet-FTRL requires $|U_t| < \infty$ in order to have finite per-round runtime.

In our setting, the set of leader mixed strategies $\mathcal{E}_t$ is deterministically determined by the context $\mathbf{z}_t$. Therefore, whenever the sequence of contexts $\{\mathbf{z}_t\}_{t \in [T]}$ is drawn from a fixed distribution, so is the sequence $\{\mathcal{E}_t\}_{t \in [T]}$, which then implies that $\{U_t\}_{t \in [T]}$ are also drawn from some fixed distribution. The last steps in order to apply logdet-FTRL are to (1) transform our action space from $[-1, 1]^K$ to the $K$-dimensional unit ball and (2) convert utilities to losses. We handle this in Algorithm 2 by rescaling our actions by $\frac{1}{\sqrt{K}}$ and negating the observed utilities before passing them to logdet-FTRL.

**Theorem 3.2.** *When $\mathcal{R}$ is instantiated as the regret minimizer of Algorithm 2, Algorithm 1 obtains expected contextual Stackelberg regret $\mathbb{E}[R(T)] = O(K^{2.5}\sqrt{T}\log(T))$ when the sequence of contexts is chosen stochastically and the sequence of follower types is chosen adversarially. The expectation is taken with respect to both the randomness in Algorithm 1, as well as the distribution over contexts.*

### 3.3 EXTENSION TO UNKNOWN UTILITIES

So far we have assumed that the leader's utility function $u$ is known. In this section, we relax this assumption and show that a modification of Algorithm 1 obtains $\tilde{O}(\sqrt{T})$ regret when $u$ is unknown, under an additional linearity assumption (Assumption 3.3).

**Assumption 3.3.** *Given context $\mathbf{z} \in \mathcal{Z}$, leader action $a_l \in \mathcal{A}_l$, and follower action $a_f \in \mathcal{A}_f$, the leader's utility is $u(\mathbf{z}, a_l, a_f) := \langle \mathbf{z}, U(a_l, a_f) \rangle$ where $U(a_l, a_f) \in \mathbb{R}^d$ is unknown to the leader.*

This setting may be thought of as both a generalization of Stackelberg games (to settings where there is side information) *and* a generalization of linear contextual bandits (to settings where another player's action influences the utility of the learner).

Our key insight is that under Assumption 3.3, the leader's utility can still be written as a linear function of some known vector $\mathbf{h}(\mathbf{z}, \mathbf{x})$, albeit in larger $(d \times K \times A_l \times A_f)$-dimensional space (Theorem 3.4). Theorem 3.4 is stated in terms of a generic distribution $\gamma$ over follower types. This distribution $\gamma$ corresponds to either the true underlying distribution over follower types $\mathbf{p}^*$ (when follower types are chosen stochastically), or the empirical distribution in hindsight over follower types (when they are chosen adversarially).

**Theorem 3.4.** *Under Assumption 3.3, the leader's expected utility (with respect to distribution $\gamma$ over follower types) can be written as $\mathbb{E}_{f \sim \gamma}[u(\mathbf{z}, \mathbf{x}, b_f(\mathbf{z}, \mathbf{x}))] = \langle \mathbf{h}(\mathbf{z}, \mathbf{x}), \boldsymbol{\theta} \rangle$ for some $\mathbf{h}(\mathbf{z}, \mathbf{x}) \in \mathbb{R}^{d \times K \times A_l \times A_f}$ which is known to the leader and $\boldsymbol{\theta} \in \mathbb{R}^{d \times K \times A_l \times A_f}$ which is not.*

The proof of Theorem 3.4 is constructive, but the closed-form expression of $\mathbf{h}(\mathbf{z}, \mathbf{x})$ is somewhat cumbersome, so we relegate it to Appendix A.

Given the results of Theorem 3.4, we can immediately obtain regret guarantees for the unknown utilities setting by running Algorithm 2 using the action set $U_t := \{\mathbf{h}(\mathbf{z}_t, \mathbf{x}) : \mathbf{x} \in \mathcal{E}_t\}$ (instead of $U_t = \{\mathbf{u}(\mathbf{z}_t, \mathbf{x}) : \mathbf{x} \in \mathcal{E}_t\}$) in round $t$. Since $\mathbf{h}(\mathbf{z}_t, \mathbf{x}) \in \mathbb{R}^{d \times K \times A_l \times A_f}$, our regret will scale as $\tilde{O}(\text{poly}(dKA_lA_f)\sqrt{T})$, compared to the $\tilde{O}(\text{poly}(K)\sqrt{T})$ rates in Section 3.1 and Section 3.2. Thus, a $\text{poly}(dA_lA_f)$ term is the price we pay for handling unknown utilities in our setting.

**Corollary 3.5.** *Under Assumption 3.3, when $\mathcal{R}$ is instantiated as the OFUL algorithm of Abbasi-Yadkori et al. (2011) and $U_t := \{\mathbf{h}(\mathbf{z}_t, \mathbf{x}) : \mathbf{x} \in \mathcal{E}_t\}$, Algorithm 1 obtains expected contextual Stackelberg regret $\mathbb{E}[R(T)] = O(dKA_lA_f\sqrt{T}\log(T))$ when the sequence of contexts is chosen adversarially and the sequence of follower types is chosen stochastically.*

**Corollary 3.6.** *Under Assumption 3.3, when $\mathcal{R}$ is instantiated as the regret minimizer of Algorithm 2 and $U_t := \{\mathbf{h}(\mathbf{z}_t, \mathbf{x}) : \mathbf{x} \in \mathcal{E}_t\}$, Algorithm 1 obtains expected contextual Stackelberg regret $\mathbb{E}[R(T)] = O((dKA_lA_f)^{2.5}\sqrt{T}\log(T))$ when the sequence of contexts is chosen stochastically and the sequence of follower types is chosen adversarially.*

### 3.4 BETTER RUNTIMES IN SPECIAL CASES

In all previous sections, the per-round runtime of Algorithm 1 is $O(\text{poly}(\mathcal{E}_t, K, A_l, A_f, d))$. In general $\mathcal{E}_t$ is exponentially-large in the size of the problem, and so the worst-case runtime of each instantiation of Algorithm 1 is exponential. This is to be expected, since we inherit the per-round NP-hardness results from the non-contextual Stackelberg game setting of Li et al. (2016), combined

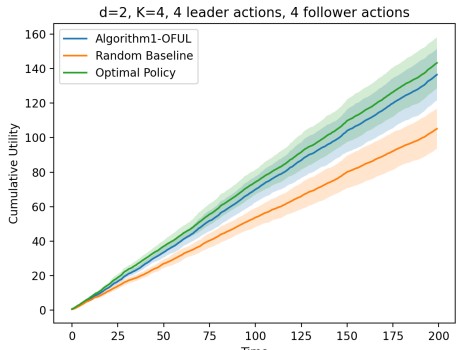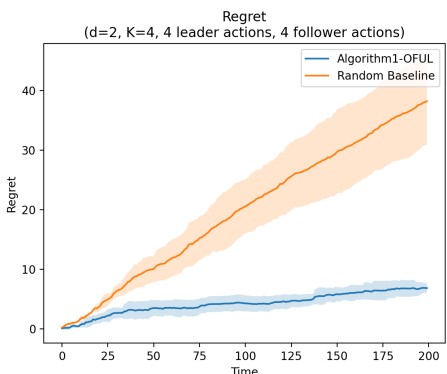

Figure 1: Left: Cumulative utility of the optimal policy, Algorithm 1 instantiated with OFUL (Alg1-OFUL), and the random baseline over $T = 200$ rounds in a setting with $4$ follower types, where each player has $4$ actions and the context dimension is also $2$. Right: Cumulative regret of Alg1-OFUL and the random baseline over $T = 200$ rounds in the same setting. Results are averaged over 10 runs.

with the offline to online reduction of Roughgarden & Wang (2019). With that being said, there are several interesting cases for which the runtime of Algorithm 1 can be improved.

**1. Small number of effective follower types** Consider a setting with three follower types, where $u_1(\mathbf{z}, \mathbf{x}, a_f)$ and $u_2(\mathbf{z}, \mathbf{x}, a_f)$ are arbitrary and $u_3(\mathbf{z}, \mathbf{x}, a_f) = \mathbb{1}\{\mathbf{z} \in \mathcal{Z}'\} \cdot u_1(\mathbf{z}, \mathbf{x}, a_f) + \mathbb{1}\{\mathbf{z} \notin \mathcal{Z}'\} \cdot u_2(\mathbf{z}, \mathbf{x}, a_f)$ for some subset of contexts $\mathcal{Z}' \subset \mathcal{Z}$. While $K = 3$, the number of approximate extreme points at each round is only $|\mathcal{E}_t| = O((2A_f^2)^{A_l} A_f^2)$, since the best-response regions of follower type 3 always overlap with those of either follower type 1 or 2. Such overlap between follower best-response regions can happen in more general settings; we capture this through the notion of *effective follower types*.

**Definition 3.7** (Effective follower types). *We say that there are $K'$ effective follower types in round $t$ if, fixing $\mathbf{z}_t$, there are $K'$ unique follower utility functions.*

When there are $K'$ effective follower types in round $t$, there are at most $|\mathcal{E}_t| = O((K'A_f^2)^{A_l} \cdot A_f^{K'})$ approximate extreme points, which may be much less than the worst-case bound of $O((KA_f^2)^{A_l} \cdot A_f^K)$ if $K'$ is small or constant.

**2. Few non-dominated leader actions per round** Similarly, it could be the case that for context $\mathbf{z}_t$, there exists two leader actions $a_l$ and $a_l'$ such that $u(\mathbf{z}_t, a_l, a_f) \leq u(\mathbf{z}_t, a_l', a_f)$ for all $a_f \in \mathcal{A}_f$. When this happens, we say that action $a_l$ is *dominated by* action $a_l'$ in round $t$. If there are $A_l'$ non-dominated actions in round $t$, then $|\mathcal{E}_t| = O((KA_f^2)^{A_l'} \cdot A_f^K)$.

**3. Exogenously-supplied leader strategies** Suppose that instead of defining $\mathcal{E}_t$ according to Definition 2.4, an external algorithm supplies a set of extreme points $\mathcal{E}_t' \subset \mathcal{E}_t$ in each round $t \in [T]$ such that $|\mathcal{E}_t'| = O(\text{poly}(A_l, A_f, K))$. If $\mathcal{E}_t' \ni \arg\max_{\mathbf{x} \in \mathcal{E}_t} \mathbb{E}[u(\mathbf{z}_t, \mathbf{x}, f_t(\mathbf{z}_t, \mathbf{x}))]$ with probability at least $1 - \delta$, then the expected regret of running Algorithm 1 using $\{\mathcal{E}_t'\}_{t=1}^T$ instead of $\{\mathcal{E}_t\}_{t=1}^T$ is $O(\mathbb{E}[R(T)] + \delta T)$, where $\mathbb{E}[R(T)]$ is the expected regret of running Algorithm 1 using $\{\mathcal{E}_t\}_{t=1}^T$.

### 3.5 EXPERIMENTS

We empirically evaluate the performance of Algorithm 1 instantiated with OFUL (henceforth Algorithm 1-OFUL) on synthetically-generated contextual Stackelberg games. In this setting, there are $4$ follower types, each of whose utility function is randomly generated. Note that since the followers' utilities depend on the contextual information, algorithms from previous work on bandit learning in Stackelberg games with side information (e.g. Algorithm 3 in Harris et al. (2024)) are not applicable in this setting. The leader's utility function is also random and is linear in the context, whose dimen-

sion is $d = 2$. Both the leader and followers have $4$ actions. Finally, both the sequence of contexts and followers are generated stochastically.

In Figure 1 we plot the cumulative utility (left) and regret (right) of Algorithm 1-OFUL and a random baseline over $T = 200$ time-steps. In the Appendix, we compare the performance of Algorithm 1-OFUL with that of Algorithm 3 in Harris et al. (2024) in the special case where follower utilities do *not* depend on the side information. Even in this setting, we find that Algorithm 1-OFUL significantly outperforms the other alternatives.

## 4    OTHER APPLICATIONS

Algorithm 1 leverages the fact that there are a finite number of follower types to transform the problem into the utility space of the leader, before applying an off-the-shelf linear contextual bandit algorithm. Interestingly, the only parts of Algorithm 1 that are specific to Stackelberg games are how the sets of extreme points and leader utilities are computed. As such, it is possible to apply Algorithm 1 to other settings where the learner has a finite number of possible utility functions. We highlight two such applications here: learning in auctions with side information (Section 4.1) and online Bayesian persuasion with side information (Section 4.2). Despite the prevalence of side information in both auctions and persuasion, we are the first to study either setting, to the best of our knowledge.

Since our definition of approximate extreme points $\mathcal{E}_t$ is specific to Stackelberg games, we instead ensure that $|U_t| < \infty$ in both settings by discretizing the *policy space*. Specifically, in both auctions and persuasion we (re-)define $\mathcal{E}_t$ to be $\{\pi^{(\boldsymbol{\omega})}(\mathbf{z}_t) \ : \ \boldsymbol{\omega} \in \Omega\}$, where $\Omega$ is a (finite) uniform grid and $\pi^{(\boldsymbol{\omega})}$ is a policy parameterized by $\boldsymbol{\omega}$. After bounding the discretization error, our analyses for the results in this section are analogous to those in Section 3.1 and Section 3.2.

### 4.1    LEARNING TO BID IN AUCTIONS WITH SIDE INFORMATION

Daskalakis & Syrgkanis (2022) consider the problem of no-regret learning in a second-price auction setting where in each round $t \in [T]$, bidders simultaneously bid on a bundle of $m$ items. Taking the perspective of a single bidder, they play bid vector $\mathbf{b}_t \in [0, 1]^m$ in round $t$ and receive the bundle of items $S(\mathbf{b}_t, \boldsymbol{\theta}_t) = \{j : \mathbf{b}_t[j] \geq \boldsymbol{\theta}_t[j]\}$, where $\boldsymbol{\theta}_t \in \Theta \subset [0, 1]^m$ is a threshold vector corresponding to the item-wise maximum of the other players' bids. Having received bundle of items $S(\mathbf{b}_t, \boldsymbol{\theta}_t)$, the bidder receives utility $u(\mathbf{b}_t, \boldsymbol{\theta}_t) := v(S(\mathbf{b}_t, \boldsymbol{\theta}_t)) - \sum_{j \in S(\mathbf{b}_t, \boldsymbol{\theta}_t)} \boldsymbol{\theta}_t[j]$, where $v(S(\mathbf{b}_t, \boldsymbol{\theta}_t)) \in \mathbb{R}$ is their *valuation* for item bundle $S(\mathbf{b}_t, \boldsymbol{\theta}_t)$ and $\sum_{j \in S(\mathbf{b}_t, \boldsymbol{\theta}_t)} \boldsymbol{\theta}_t[j]$ is the cumulative price of the items in $S(\mathbf{b}_t, \boldsymbol{\theta}_t)$. Daskalakis & Syrgkanis (2022) provide a no-regret learning algorithm for this setting when each threshold vector $\boldsymbol{\theta}_t$ can take only one of $K$ different values (i.e. $|\Theta| = K$). In the bandit feedback setting, the threshold vector $\boldsymbol{\theta}_t$ is never revealed to the learner.

We apply a slightly more general version of Algorithm 1 (Algorithm 3) to a generalization of this problem, where the bidder's valuation is allowed to depend on additional contextual information (i.e. $v : \mathcal{Z} \times [0, 1]^m \times \Theta \to \mathbb{R}$). Such contextual information is often present in auction settings. For example, shoppers' valuations for bundles of clothing items often depend on external factors such as the season or current fashion trends.

In this setting, utilities are now a function of the context $\mathbf{z}_t$, the bid vector $\mathbf{b}_t$, and the threshold vector $\boldsymbol{\theta}_t$ and a policy is a mapping from contexts to bids for each item (i.e. $\pi : \mathcal{Z} \to [0, 1]^m$). Instead of discretizing the learner's action space like in Section 3, we instead discretize their policy space as follows.

**Definition 4.1** (Discretized Policy for Auctions). *Let* $\Omega := \{\boldsymbol{\omega} \in \Delta^K, \ T \cdot \boldsymbol{\omega}[i] \in \mathbb{N}, \ \forall i \in [K]\}$. *We define policy* $\pi^{(\boldsymbol{\omega})}$ *as* $\pi^{(\boldsymbol{\omega})}(\mathbf{z}) \ := \ \arg\max_{\mathbf{b} \in [0,1]^m} \sum_{i=1}^{K} \boldsymbol{\omega}[i] \ \cdot \ u(\mathbf{z}, \mathbf{b}, \boldsymbol{\theta}^{(i)})$ *and* $\mathcal{E}_t := \{\pi^{(\boldsymbol{\omega})}(\mathbf{z}_t) \ : \ \boldsymbol{\omega} \in \Omega\}$.

Armed with this policy discretization, we are ready to state our results for running Algorithm 3 in repeated auctions with side information. Analogous to Definition 2.1, we define regret to be the cumulative difference in utility between the optimal policy and the sequence of bid vectors played by the learner.

**Corollary 4.2.** *When $U_t := \{\mathbf{u}(\mathbf{z}_t, \mathbf{b}) \ : \ \mathbf{b} \in \mathcal{E}_t\}$ and $\mathcal{R}$ is instantiated as the OFUL algorithm of Abbasi-Yadkori et al. (2011), the expected regret of Algorithm 3 is $\mathbb{E}[R(T)] = O(K\sqrt{T}\log(T))$ when the sequence of contexts is chosen adversarially and the sequence of threshold vectors is chosen stochastically.*

**Corollary 4.3.** *When $U_t := \{\mathbf{u}(\mathbf{z}_t, \mathbf{b}) \ : \ \mathbf{b} \in \mathcal{E}_t\}$ and $\mathcal{R}$ is instantiated as the regret minimizer of Algorithm 2, Algorithm 3 obtains expected regret $\mathbb{E}[R(T)] = O(K^{2.5}\sqrt{T}\log(T))$ when the sequence of contexts is chosen stochastically and the sequence of threshold vectors is chosen adversarially.*

## 4.2 BAYESIAN PERSUASION WITH PUBLIC AND PRIVATE STATES

Bayesian persuasion (BP; Kamenica & Gentzkow (2011)) is a canonical setting in information design which studies how provision of information by an informed designer (the *sender*) influences the strategic behavior of agents (*receivers*) in a game.

We study a generalization of the *online* BP setting, in which a sender learns to play against a sequence of $T$ receivers, which was first introduced by Castiglioni et al. (2020). The novelty in our setting is that a context $\mathbf{z}_t \in \mathcal{Z}$ is revealed to both the sender and receiver in each round $t \in [T]$. This context may be thought of as a "public state", which contains contextual information that is available to both players. After observing the context, the sender commits to a *signaling policy* $\mu : \Omega \to \mathcal{A}$, which maps *private* states from some finite set $\Omega$ to receiver actions in finite set $\mathcal{A}$.[5]The private state is drawn from a publicly-known prior distribution and revealed to the sender (but not the receiver). After the private state is realized, the sender signals according to their policy and the follower takes an action (possibly different from the one recommended to them by the sender).

The sender faces a sequence of receivers $r_1, \ldots, r_T$, where each receiver $r_t$ is one of $K$ types $\{\tau^{(1)}, \ldots, \tau^{(K)}\}$. Our notion of receiver type is analogous to our definition of follower type in Section 2, i.e. each receiver type has a different utility function which maps contexts, private states, and receiver actions to utilities. As is standard in most BP settings, we assume that receivers are Bayes-rational and pick their action to maximize their expected utility with respect to the posterior distribution over states induced by the sender's signal realization.

It is possible to show that the set of leader signaling policies can be represented by a convex polytope $\mathcal{P}$ (see, e.g. Section 4 in Bernasconi et al. (2023)). As such, the leader can solve for the optimal signaling policy to play given a context $\mathbf{z}_t$ and distribution over receiver types by optimizing over $\mathcal{P}$. The leader's goal is to maximize their own cumulative utility $u : \mathcal{Z} \times \mathcal{P} \times \{\tau^{(1)}, \ldots \tau^{(K)}\} \to [-1, 1]$, which is a function of the context (i.e. public state), the private state, and the receiver's type (through the action they take). Under bandit feedback, the sequence of receiver types $r_1, \ldots, r_T$ is never revealed to the sender.

We discretize the policy space analogously to Section 4.1; the only difference is the form of the leader's utility function and the action space they are optimizing over.

**Definition 4.4** (Discretized Policy for Persuasion). *Let $\Omega := \{\boldsymbol{\omega} \in \Delta^K, \ T \cdot \boldsymbol{\omega}[i] \in \mathbb{N}, \ \forall i \in [K]\}$. We define policy $\pi^{(\boldsymbol{\omega})}$ as $\pi^{(\boldsymbol{\omega})}(\mathbf{z}) := \arg\max_{\mu \in \mathcal{P}} \sum_{i=1}^{K} \boldsymbol{\omega}[i] \cdot u(\mathbf{z}, \mu, \tau^{(i)})$ and $\mathcal{E}_t := \{\pi^{(\boldsymbol{\omega})}(\mathbf{z}_t) \ : \ \boldsymbol{\omega} \in \Omega\}$.*

We obtain results for two persuasion settings with side information: one in which the sequence of public states is chosen adversarially and the receiver types are chosen stochastically, and one where the sequence of contexts is stochastic and the follower types are chosen stochastically.

**Corollary 4.5.** *When $U_t := \{\mathbf{u}(\mathbf{z}_t, \mu) \ : \ \mu \in \mathcal{E}_t\}$ and $\mathcal{R}$ is instantiated as the OFUL algorithm of Abbasi-Yadkori et al. (2011), the expected regret of Algorithm 1 is $\mathbb{E}[R(T)] = O(K\sqrt{T}\log(T))$ when the sequence of public states is chosen adversarially and the sequence of receiver types is chosen stochastically.*

**Corollary 4.6.** *When $U_t := \{\mathbf{u}(\mathbf{z}_t, \mu) \ : \ \mu \in \mathcal{E}_t\}$ and $\mathcal{R}$ is instantiated as the regret minimizer of Algorithm 2, Algorithm 1 obtains expected regret $\mathbb{E}[R(T)] = O(K^{2.5}\sqrt{T}\log(T))$ when the*

---

[5]This is without loss of generality due to a revelation principle-style argument (see, e.g. Kamenica & Gentzkow (2011)).

*sequence of public states is chosen stochastically and the sequence of receiver types is chosen adversarially.*

## 5 CONCLUSION

We study the problem of bandit learning in Stackelberg games with side information, where we improve upon the previously best-known $\tilde{O}(T^{2/3})$ regret rates to $\tilde{O}(T^{1/2})$. Our results rely on a reduction to linear contextual bandits in the leader's utility space. Extensions to unknown leader utilities, auctions with side information, and Bayesian persuasion with public and private states are also considered.

There are several exciting directions for future work. While our results for known utilities extend to auctions and persuasion, our results for unknown utilities do not. It would be interesting to see if Algorithm 1 can be (further) generalized to handle such settings. Given the exponential worst-case computational complexity of Algorithm 1, a more in depth study of its runtime using tools from, e.g. smoothed analysis Spielman & Teng (2004) would also be interesting.

## ACKNOWLEDGMENTS

We would like to thank the anonymous reviewers for their helpful feedback. For part of this work, KH was a Ph.D. student at Carnegie Mellon University. This work was supported in part by NSF Grant CCF-1910321, an ERC grant (Project 101165466 — PLA-STEER), the FAIR (Future Artificial Intelligence Research) project PE0000013, and the EU Horizon project ELIAS (No. 101120237).

The work of Martino Bernasconi and AC was partially funded by the European Union. Views and opinions expressed are however those of the author(s) only and do not necessarily reflect those of the European Union or the European Research Council Executive Agency. Neither the European Union nor the granting authority can be held responsible for them.

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

# A  APPENDIX FOR SECTION 3: A REDUCTION TO LINEAR CONTEXTUAL BANDITS

---

**Algorithm 2:** Regret Minimizer $\tilde{\mathcal{R}}$

---

Let $\mathcal{R}'$ be logdet-FTRL (Algorithm 1 of Liu et al. (2023))
`Recommend(`$U_t$`):`
**begin**

    Create scaled action set $U_t' = \left\{ \frac{\mathbf{v}}{\sqrt{K}} : \mathbf{v} \in U_t \right\}$;

    $\mathbf{v}_t' = \mathcal{R}'.\text{recommend}(U_t')$

    **return** $\sqrt{K} \cdot \mathbf{v}_t'$;

**end**
`ObserveUtility(`$\mathbf{v}_t$`, `$u_t$`):`
**begin**

    Set $\mathbf{v}_t' = \frac{\mathbf{v}_t}{\sqrt{K}}$ and $u_t' = -\frac{u_t}{\sqrt{K}}$;

    Call $\mathcal{R}'.\text{observeLoss}(\mathbf{v}_t', u_t')$;

**end**

---

**Theorem 3.1.** *When $\mathcal{R}$ is instantiated as the OFUL algorithm of Abbasi-Yadkori et al. (2011), Algorithm 1 obtains expected contextual Stackelberg regret $\mathbb{E}[R(T)] = O(K\sqrt{T}\log(T))$ when the sequence of contexts is chosen adversarially and the sequence of follower types is chosen stochastically. The expectation is taken with respect to both the randomness in Algorithm 1, as well as the distribution over follower types.*

*Proof.* Let $\mathbf{p}^* \in \Delta(K)$ be the distribution over follower types. Define $\mathbf{u}(\mathbf{z}, \mathbf{x})$ is a vector in $\mathbb{R}^K$ where for each $k \in [K]$:
$$\mathbf{u}(\mathbf{z}, \mathbf{x})[k] = u(\mathbf{z}, \mathbf{x}, b_k(\mathbf{z}, \mathbf{x}))$$
Observe that for a fixed $\mathbf{z}, \mathbf{x}$, we have that
$$u(\mathbf{z}, \mathbf{x}, b_{f_t}(\mathbf{z}, \mathbf{x})) = \langle \mathbf{p}^*, \mathbf{u}(\mathbf{z}, \mathbf{x}) \rangle + (u(\mathbf{z}, \mathbf{x}, b_{f_t}(\mathbf{z}, \mathbf{x})) - \langle \mathbf{p}^*, \mathbf{u}(\mathbf{z}, \mathbf{x}) \rangle)$$
Let $\eta_t := u(\mathbf{z}, \mathbf{x}, b_{f_t}(\mathbf{z}, \mathbf{x})) - \langle \mathbf{p}^*, \mathbf{u}(\mathbf{z}, \mathbf{x}) \rangle$. Observe that since $\mathbb{E}[u(\mathbf{z}, \mathbf{x}, b_{f_t}(\mathbf{z}, \mathbf{x}))] = \langle \mathbf{p}^*, \mathbf{u}(\mathbf{z}, \mathbf{x}) \rangle$, $\eta_t$ is a zero-mean random variable bounded in $[-2, 2]$. Similarly we have that for $a_l \sim \mathbf{x}$,
$$u(\mathbf{z}, a_l, b_{f_t}(\mathbf{z}, \mathbf{x})) = u(\mathbf{z}, \mathbf{x}, b_{f_t}(\mathbf{z}, \mathbf{x})) + (u(\mathbf{z}, a_l, b_{f_t}(\mathbf{z}, \mathbf{x})) - u(\mathbf{z}, \mathbf{x}, b_{f_t}(\mathbf{z}, \mathbf{x}))),$$
where $\gamma_t := u(\mathbf{z}, a_l, b_{f_t}(\mathbf{z}, \mathbf{x})) - u(\mathbf{z}, \mathbf{x}, b_{f_t}(\mathbf{z}, \mathbf{x}))$ is a zero-mean random variable bounded in $[-2, 2]$. Putting both terms together, we have that
$$u(\mathbf{z}, a_l, b_{f_t}(\mathbf{z}, \mathbf{x})) = \langle \mathbf{p}^*, \mathbf{u}(\mathbf{z}, \mathbf{x}) \rangle + \epsilon_t,$$
where $\epsilon_t := \eta_t + \gamma_t$ is a zero-mean random variable bounded in $[-4, 4]$.

$$\mathbb{E}[R(T)] = \mathbb{E}_{f_1, \dots, f_T} \left[ \sum_{t=1}^{T} u(\mathbf{z}_t, \pi^*(\mathbf{z}_t), b_{f_t}(\mathbf{z}_t, \pi^*(\mathbf{z}_t))) - u(\mathbf{z}_t, \mathbf{x}_t, b_{f_t}(\mathbf{z}_t, \mathbf{x}_t)) \right]$$

$$\leq 1 + \sum_{t=1}^{T} \mathbb{E}_{f_1, \dots, f_t} [u(\mathbf{z}_t, \pi^{(\mathcal{E})}(\mathbf{z}_t), b_{f_t}(\mathbf{z}_t, \pi^{(\mathcal{E})}(\mathbf{z}_t))) - u(\mathbf{z}_t, \mathbf{x}_t, b_{f_t}(\mathbf{z}_t, \mathbf{x}_t))]$$

$$= 1 + \sum_{t=1}^{T} \mathbb{E}_{f_1, \dots, f_{t-1}} [\mathbb{E}_t[u(\mathbf{z}_t, \pi^{(\mathcal{E})}(\mathbf{z}_t), b_{f_t}(\mathbf{z}_t, \pi^{(\mathcal{E})}(\mathbf{z}_t)))] - \mathbb{E}_t[u(\mathbf{z}_t, \mathbf{x}_t, b_{f_t}(\mathbf{z}_t, \mathbf{x}_t))]]$$

$$= 1 + \sum_{t=1}^{T} \langle \mathbf{p}^*, \mathbf{u}(\mathbf{z}_t, \pi^{(\mathcal{E})}(\mathbf{z}_t)) \rangle - \langle \mathbf{p}^*, \mathbb{E}_{f_1, \dots, f_{t-1}}[\mathbf{u}(\mathbf{z}_t, \mathbf{x}_t)] \rangle$$

$$= 1 + \mathbb{E}_{f_1, \dots, f_T} \left[ \sum_{t=1}^{T} \langle \mathbf{p}^*, \mathbf{u}(\mathbf{z}_t, \pi^{(\mathcal{E})}(\mathbf{z}_t)) \rangle - \langle \mathbf{p}^*, \mathbf{u}(\mathbf{z}_t, \mathbf{x}_t) \rangle \right]$$

$$\leq 2 + 4\sqrt{TK\log(\lambda + T)} (\sqrt{\lambda}K + 4\sqrt{2\log(T) + K\log(1 + T/\lambda)})$$

where $\pi^{(\mathcal{E})}$ is the optimal policy which is restricted to $\mathcal{E}_t$ in round $t$, the second line follows from Lemma 4.4 in Harris et al. (2024) and the last line follows from applying the regret guarantee of Algorithm 1 in Abbasi-Yadkori et al. (2011). $\qquad\square$

**Theorem 3.2.** *When $\mathcal{R}$ is instantiated as the regret minimizer of Algorithm 2, Algorithm 1 obtains expected contextual Stackelberg regret $\mathbb{E}[R(T)] = O(K^{2.5}\sqrt{T}\log(T))$ when the sequence of contexts is chosen stochastically and the sequence of follower types is chosen adversarially. The expectation is taken with respect to both the randomness in Algorithm 1, as well as the distribution over contexts.*

*Proof.*

$$\mathbb{E}[R(T)] = \mathbb{E}_{\mathbf{z}_1,\ldots,\mathbf{z}_T}[\sum_{t=1}^{T} u(\mathbf{z}_t, \pi^*(\mathbf{z}_t), b_{f_t}(\mathbf{z}_t, \pi^*(\mathbf{z}_t))) - u(\mathbf{z}_t, \mathbf{x}_t, b_{f_t}(\mathbf{z}_t, \mathbf{x}_t))]$$

$$\leq 1 + \mathbb{E}_{\mathbf{z}_1,\ldots,\mathbf{z}_T}[\sum_{t=1}^{T} u(\mathbf{z}_t, \pi^{(\mathcal{E})}(\mathbf{z}_t), b_{f_t}(\mathbf{z}_t, \pi^{(\mathcal{E})}(\mathbf{z}_t))) - u(\mathbf{z}_t, \mathbf{x}_t, b_{f_t}(\mathbf{z}_t, \mathbf{x}_t))]$$

$$= 1 + \mathbb{E}_{\mathbf{z}_1,\ldots,\mathbf{z}_T}[\sum_{t=1}^{T} \langle \mathbf{u}(\mathbf{z}_t, \pi^{(\mathcal{E})}(\mathbf{z}_t)), \mathbf{1}_{f_t}\rangle - \langle \mathbf{u}(\mathbf{z}_t, \mathbf{x}_t), \mathbf{1}_{f_t}\rangle]$$

$$= 1 + \mathbb{E}_{\mathbf{z}_1,\ldots,\mathbf{z}_T}[\sum_{t=1}^{T} \langle \mathbf{u}(\mathbf{z}_t, \pi^{(\mathcal{E})}(\mathbf{z}_t)), \mathbf{1}_{f_t}\rangle - \langle \mathbf{v}_t, \mathbf{1}_{f_t}\rangle]$$

$$= 1 + \mathbb{E}_{\mathbf{z}_1,\ldots,\mathbf{z}_T}[\sum_{t=1}^{T} \langle \tilde{\pi}(U_t), \mathbf{1}_{f_t}\rangle - \langle \mathbf{v}_t, \mathbf{1}_{f_t}\rangle]$$

$$= 1 + \sqrt{K} \cdot \mathbb{E}_{\mathbf{z}_1,\ldots,\mathbf{z}_T}[\sum_{t=1}^{T} \langle \frac{\tilde{\pi}(U_t)}{\sqrt{K}}, \mathbf{1}_{f_t}\rangle - \langle \frac{\mathbf{v}_t}{\sqrt{K}}, \mathbf{1}_{f_t}\rangle]$$

$$= O(K^{2.5}\sqrt{T}\log(T))$$

where $\pi^{(\mathcal{E})}$ is the optimal policy which is restricted to $\mathcal{E}_t$ in round $t$, $\tilde{\pi}(U_t) := \mathbf{u}(\mathbf{z}_t, \pi^{(\mathcal{E})}(\mathbf{z}_t))$, the second line follows from Lemma 4.4 in Harris et al. (2024) and the last line follows from the regret guarantee of Algorithm 1 in Liu et al. (2023). To apply this result, we use the fact that the $K$-dimensional unit cube with side length 2 is contained in the $K$-dimensional unit ball with radius $\sqrt{K}$. $\qquad\square$

**Theorem 3.4.** *Under Assumption 3.3, the leader's expected utility (with respect to distribution $\gamma$ over follower types) can be written as $\mathbb{E}_{f\sim\gamma}[u(\mathbf{z}, \mathbf{x}, b_f(\mathbf{z}, \mathbf{x}))] = \langle \mathbf{h}(\mathbf{z}, \mathbf{x}), \boldsymbol{\theta}\rangle$ for some $\mathbf{h}(\mathbf{z}, \mathbf{x}) \in \mathbb{R}^{d\times K\times A_l\times A_f}$ which is known to the leader and $\boldsymbol{\theta} \in \mathbb{R}^{d\times K\times A_l\times A_f}$ which is not.*

*Proof.*

$$\mathbb{E}_{f\sim\gamma}[u(\mathbf{z}, \mathbf{x}, b_f(\mathbf{z}, \mathbf{x}))] = \sum_{i=1}^{K} u(\mathbf{z}, \mathbf{x}, b_i(\mathbf{z}, \mathbf{x}))\mathbb{P}_\gamma(f = i)$$

$$= \sum_{i=1}^{K} \sum_{a_l\in\mathcal{A}_l} \sum_{a_f\in\mathcal{A}_f} \mathbf{z}^\top\mathbf{x}[a_l]\mathbb{1}\{a_f = b_i(\mathbf{z}, \mathbf{x})\}U(a_l, a_f)\mathbb{P}_\gamma(f = i)$$

Let $i \in [K]$, $a_l \in \mathcal{A}_f$, $a_f \in \mathcal{A}_f$, and $j \in [d]$. Define
$$n(i, a_l, a_f, j) := (i-1)\cdot(A_l\cdot A_f\cdot d) + (a_l-1)\cdot(A_f\cdot d) + (a_f-1)\cdot d + j$$

Let $\boldsymbol{\theta}_{i,a_l,a_f} := U(a_l, a_f)\mathbb{P}_\gamma(f = i) \in \mathbb{R}^d$ and define $\boldsymbol{\theta} \in \mathbb{R}^{d\times K\times A_l\times A_f}$ such that
$$\boldsymbol{\theta}[n(i, a_l, a_f, j)] := \boldsymbol{\theta}_{i,a_l,a_f}[j].$$

Similarly, let
$$\mathbf{h}(\mathbf{z}, \mathbf{x})[n(i, a_l, a_f, j)] := \mathbf{z}[j]\mathbf{x}[a_l]\mathbb{1}\{a_f = b_i(\mathbf{z}, \mathbf{x})\}.$$

$\qquad\square$

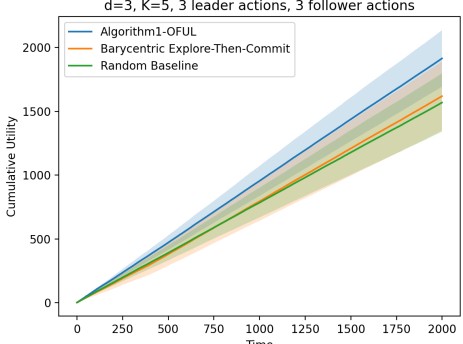 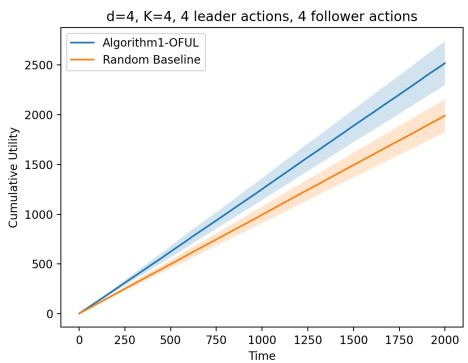

(a) Cumulative utility of Algorithm 1 instantiated with OFUL (Algorithm1-OFUL), Algorithm 3 of Harris et al. (2024), and the random baseline over $T = 2,000$ rounds in a setting with 5 follower types, where each player has 3 actions and the context dimension is also 3. Results are averaged over 10 runs. The hyperparameter of Algorithm 3 of Harris et al. (2024) was tuned to maximize performance.

(b) Cumulative utility of Algorithm 1 instantiated with OFUL (Algorithm1-OFUL) and the random baseline over $T = 2,000$ rounds in a setting with 4 follower types, where each player has 4 actions and the context dimension is also 4. Results are averaged over 10 runs. Algorithm 3 of Harris et al. (2024) is not applicable in this setting because the follower's utility depends on the context.

Figure 2: Additional empirical results.

## A.1 APPENDIX FOR SECTION 3.5: EXPERIMENTS

Here we compare to Algorithm 3 in Harris et al. (2024) (henceforth Barycentric Explore-Then-Commit), which obtains $\tilde{O}(T^{2/3})$ regret in the special case where the follower's utility does not depend on the context. At a high level, Barycentric Explore-Then-Commit repeatedly plays a small number of leader mixed strategies to estimate the frequency of follower best-responses, before acting greedily with respect to these estimates for the remaining rounds. We also compare both algorithms to a baseline which plays by sampling leader mixed strategies uniformly-at-random in each round (henceforth Random Baseline).

In Figure 2a, we compare the performance of the three algorithms on synthetic data. There are 5 follower types, each of whose utility function is randomly generated and does *not* depend on the contextual information. The leader's utility function is also random and is linear in the context, whose dimension is $d = 3$. Both the leader and followers have 3 actions. Finally, both the sequence of contexts and followers are generated stochastically.

In Figure 2b, we compare the performance of Algorithm 1-OFUL with that of Random Baseline in a setting where follower utilities *do* depend on contextual information. As a result, Barycentric Explore-Then-Commit is not applicable. In this setting, both leader and follower utility functions are random linear functions of the context player actions. $d = K = 4$, and both players have 4 actions. We find that in both settings Algorithm 1-OFUL significantly outperforms Random Baseline and Barycentric Explore-Then-Commit (where applicable).

## B APPENDIX FOR SECTION 4: OTHER APPLICATIONS

**Corollary 4.2.** *When $U_t := \{\mathbf{u}(\mathbf{z}_t, \mathbf{b}) \; : \; \mathbf{b} \in \mathcal{E}_t\}$ and $\mathcal{R}$ is instantiated as the OFUL algorithm of Abbasi-Yadkori et al. (2011), the expected regret of Algorithm 3 is $\mathbb{E}[R(T)] = O(K\sqrt{T}\log(T))$ when the sequence of contexts is chosen adversarially and the sequence of threshold vectors is chosen stochastically.*

*Proof.* Observe that

$$\pi^*(\mathbf{z}) = \arg\max_{\mathbf{b}\in[0,1]^m} \mathbb{E}_{\boldsymbol{\theta}\sim\mathcal{P}}[u(\mathbf{z}, \mathbf{b}, \boldsymbol{\theta})] = \arg\max_{\mathbf{b}\in[0,1]^m} \sum_{i=1}^{K} \mathbf{p}[i] \cdot u(\mathbf{z}, \mathbf{b}, \boldsymbol{\theta}^{(i)}),$$

---

**Algorithm 3:** Reduction for Auctions and Persuasion

---

**Input:** Linear contextual bandit algorithm $\mathcal{R}$

**for** $t = 1, \ldots, T$ **do**

    Observe $\mathbf{z}_t$, compute utility set $U_t$

    Let $\mathbf{v}_t \leftarrow \mathcal{R}.\text{recommend}(U_t)$

    Play the action which induces $\mathbf{v}_t$

    Receive utility $u_t$ and call $\mathcal{R}.\text{observeUtility}(\mathbf{v}_t, u_t)$

**end**

---

where $\mathcal{P}$ is an unknown distribution with support on $\{\boldsymbol{\theta}^{(1)}, \ldots, \boldsymbol{\theta}^{(K)}\}$. Let $\pi' := \pi^{(\boldsymbol{\omega})}$ be the optimal policy in the discretization and let $\mathcal{P}'$ be the corresponding distribution over $\boldsymbol{\theta}$. We have that

$$R(T) = \mathbb{E}_{\boldsymbol{\theta} \sim \mathcal{P}}[\sum_{t=1}^{T} u(\mathbf{z}_t, \pi^*(\mathbf{z}_t), \boldsymbol{\theta}) - u(\mathbf{z}_t, \mathbf{b}_t, \boldsymbol{\theta})]$$

$$= \sum_{t=1}^{T} \left( \sum_{i=1}^{K} \mathbf{p}'[i](u(\mathbf{z}_t, \pi'(\mathbf{z}_t), \boldsymbol{\theta}^{(i)}) - u(\mathbf{z}_t, \mathbf{b}_t, \boldsymbol{\theta}^{(i)})) + \sum_{i=1}^{K} (\mathbf{p}'[i] - \mathbf{p}[i]) \cdot u(\mathbf{z}_t, \mathbf{b}_t, \boldsymbol{\theta}^{(i)}) \right.$$

$$\left. + \sum_{i=1}^{K} (\mathbf{p}[i] - \mathbf{p}'[i]) \cdot u(\mathbf{z}_t, \pi^*(\mathbf{z}_t), \boldsymbol{\theta}^{(i)}) + \sum_{i=1}^{K} \mathbf{p}'[i](u(\mathbf{z}_t, \pi^*(\mathbf{z}_t), \boldsymbol{\theta}^{(i)}) - u(\mathbf{z}_t, \pi'(\mathbf{z}_t), \boldsymbol{\theta}^{(i)})) \right)$$

$$\leq 2K + \mathbb{E}_{\boldsymbol{\theta}_1, \ldots, \boldsymbol{\theta}_T \sim \mathcal{P}'} \left[ \sum_{t=1}^{T} u(\mathbf{z}_t, \pi'(\mathbf{z}_t), \boldsymbol{\theta}_t) - u(\mathbf{z}_t, \mathbf{b}_t, \boldsymbol{\theta}_t) \right]$$

The rest of the proof follows identically to the proof of Theorem 3.1, but without the discretization step. $\qquad\square$

**Corollary 4.3.** *When $U_t := \{\mathbf{u}(\mathbf{z}_t, \mathbf{b}) \ : \ \mathbf{b} \in \mathcal{E}_t\}$ and $\mathcal{R}$ is instantiated as the regret minimizer of Algorithm 2, Algorithm 3 obtains expected regret $\mathbb{E}[R(T)] = O(K^{2.5}\sqrt{T}\log(T))$ when the sequence of contexts is chosen stochastically and the sequence of threshold vectors is chosen adversarially.*

*Proof.* The proof follows identically to the proof of Theorem 3.2, but without the loss in utility due to discretization. To see why, let $N_i := \sum_{t:\boldsymbol{\theta}_t = \boldsymbol{\theta}^{(i)}} 1$ and observe that

$$\pi^*(\mathbf{z}) := \arg \max_{\mathbf{b} \in [0,1]^m} \mathbb{E}_{\mathbf{z} \sim \mathcal{P}}[\sum_{t=1}^{T} u(\mathbf{z}, \mathbf{b}, \boldsymbol{\theta}_t)]$$

$$= \arg \max_{\mathbf{b} \in [0,1]^m} \sum_{i=1}^{K} \frac{N_i}{T} \cdot \mathbb{E}_{\mathbf{z} \sim \mathcal{P}}[u(\mathbf{z}, \mathbf{b}, \boldsymbol{\theta}^{(i)})]$$

$\qquad\square$

