# OpenReview forum: "Nearly-Optimal Bandit Learning in Stackelberg Games with Side Information"
_ICLR.cc/2026/Conference — ICLR 2026 Poster_

### Official Review · Reviewer_dzah · 2025-10-26

**Soundness:** 3
**Presentation:** 3
**Contribution:** 2
**Rating:** 4
**Confidence:** 4

**Summary:**

This paper studies online learning for contextual Stackelberg games under bandit feedback. This paper uses a reduction that lets the leader act in a utility space and then plug in linear contextual bandits algorithms. Instantiating the reduction with OFUL gives $\tilde{O}(\sqrt{T})$ regret when contexts are adversarial and follower types are stochastic; a logdet-FTRL instantiation gives $\tilde{O}\left(K^{2.5}\sqrt{T}\right)$ regret when contexts are stochastic and follower types are adversarial. The paper also extends to unknown leader utilities  with $\tilde{O}\bigl(\mathrm{poly}(d,K,A_\ell,A_f)\sqrt{T}\bigr)$ regret.

**Strengths:**

A clean reduction that operates in the leader’s utility space, enabling off-the-shelf contextual bandit algorithms to close the bandit-feedback gap to $\tilde{O}(\sqrt{T})$, with extensions to unknown utilities and applications such as auctions and Bayesian persuasion.

**Weaknesses:**

1. The paper restricts the leader’s strategy space to the set of approximate extreme points of contextual best-response regions. It would be more interesting to allow approximate follower responses, i.e., followers who play $\delta$-suboptimal best responses, and analyze how such deviations affect the reduction and regret guarantees.
2.  The main results rely heavily on prior techniques—off-the-shelf linear contextual bandit algorithms, the Contextual Follower Best-Response Region framework, and $\delta$-approximate extreme points—so the paper’s technical novelty is unclear; overall, the contribution appears incremental, primarily composing existing components.

**Questions:**

No specific questions

---

> ### Author Response · Authors · 2025-11-19
>
> Thanks for taking the time to review our submission. Our responses to your comments are below.
>
> > *The paper restricts the leader’s strategy space to the set of approximate extreme points of contextual best-response regions. It would be more interesting to allow approximate follower responses, i.e., followers who play suboptimal best responses, and analyze how such deviations affect the reduction and regret guarantees.*
>
> Regarding our use of approximate extreme points, this is (nearly) without loss of generality, as restricting ourselves to this set only results in adding at most a constant term to the leader’s regret. (For more details, see our response to Reviewer mHpn.)
>
> Regarding allowing the follower to approximately best-respond, we did not consider this as our main focus was closing the O(T^{⅔}) to O(T^{1/2}) regret gap in our setting, which was an open problem in the literature posed by Harris et al., NeurIPS 2024. However, we agree that this would be an interesting relaxation to study in future research.
>
> > *The main results rely heavily on prior techniques—off-the-shelf linear contextual bandit algorithms, the Contextual Follower Best-Response Region framework, and delta-approximate extreme points—so the paper’s technical novelty is unclear; overall, the contribution appears incremental, primarily composing existing components.*
>
> As we inherit the contextual Stackelberg game setting of Harris et al., we do make use of some of their machinery for discretizing the leader’s mixed strategy space (contextual follower best-response regions, delta-approximate extreme points).
> However, we view our reduction to off-the-shelf contextual bandit algorithms as a strength of our work, not a weakness. By leveraging this reduction and finding two specific contextual bandit algorithms which are applicable, we are able to close the regret gap from O(T^{2/3}) to O(T^{1/2}) under bandit feedback for the two settings of interest (adversarial (stochastic) contexts and stochastic (adversarial) follower types).
>
> Moreover, the generic form of our reduction allows for the possibility of new contextual bandit algorithms to be “plugged in” to obtain new results in our setting. As one example, if the logdet-FTRL algorithm of Liu et al. were improved upon, it would automatically give us better rates in K (# of follower types) for the adversarial follower type regime. As another example, if someone were to provide a contextual bandit algorithm for a setting where some *bounded* adversary can *jointly* select the context and the learner’s action set at each round, then this algorithm could be used as a part of our reduction to obtain results in our setting when a bounded adversary jointly selects the sequence of contexts and follower types.

---

### Official Review · Reviewer_mHpn · 2025-10-29

**Soundness:** 3
**Presentation:** 3
**Contribution:** 3
**Rating:** 6
**Confidence:** 3

**Summary:**

This paper studies contextual Stackelberg games with multiple follower types under bandit feedback. The authors propose a reduction to linear contextual bandits in the leader’s utility space, enabling near-optimal learning algorithms for two settings: (i) adversarial contexts with stochastic followers, and (ii) stochastic contexts with adversarial followers. In both cases, the algorithm achieves $\tilde{\mathcal{O}}(T^{1/2})$ regret, improving upon the previous best-known $\tilde{\mathcal{O}}(T^{2/3})$ rate. The framework also extends to second-price auctions and Bayesian persuasion with side information through discretized action spaces.

**Strengths:**

- This work closes the gap between lower and upper regret bounds for bandit-feedback contextual Stackelberg games, achieving the optimal rate up to logarithmic factors.

- The reduction to linear contextual bandits in utility space is conceptually clean and broadly applicable as shown by examples.

- Extension to unknown utilities: Maintains near-optimal regret under a linearity assumption, showing robustness of the reduction.

- The paper is well structured and the main theorems and algorithmic structure are well organized and technically sound.

**Weaknesses:**

- The per-round time of the proposed algorithm is exponential which limits the practicability of the algorithm, which is also evident from the small toy example of Sec. 3.5.

- The authors assume a finite and known set of follower types and full knowledge of their utility functions (except in the linear “unknown utility” variant).

- The discretization of the leader’s strategy space $\mathcal{E}_t$ is important to keep the action space finite but introduces an approximation error on the regret that is only bounded asymptotically.

**Questions:**

1) In limitations, it is mentioned that a possible path forward towards reducing the runtime by means of smoothed analysis. Could the authors comment on the conceptual idea and how promising might be?

2) How restrictive is the leader assumptions on the followers, i.e., cardinality, types, and utility, in practice? Do the authors see any path forward at relaxing these assumptions (cardinality and types)?

3) Is it possible to say antyhing about the $\mathcal{O}(1)$ approximation error in the regret due to the discretization?

---

> ### Author Response · Authors · 2025-11-19
>
> Thanks for taking the time to review our submission. Please find our replies to your questions and comments below.
>
> > *In limitations, it is mentioned that a possible path forward towards reducing the runtime by means of smoothed analysis. Could the authors comment on the conceptual idea and how promising might be?*
>
> Thanks for the interesting question! We would like to point out that exponential runtime is unavoidable in the worst case, due to existing lower bounds for the non-contextual setting (see our submission for references). With that being said, our algorithm’s exponential runtime is due to its dependence on an exponentially-large number of extreme points of different “best-response regions”, which discretize the leader’s mixed strategy space in a nearly-lossless way (more on this later). If we could find a polynomially-sized set of mixed strategies that is guaranteed to contain the optimal strategy (for the given context), then our reduction would run in polynomial time.
>
> Smoothed analysis asks what happens when an algorithm’s inputs are subjected to random perturbations. In our setting, it would be interesting to see if there are certain groupings of follower types that allow the leader to get away with considering a small (i.e. polynomial) number of extreme points. The geometry of the leader’s best-response regions is highly sensitive to perturbations, and so some sort of smoothing may eliminate the worst cases which lead to an exponential blow-up in the size of the set of extreme points.
>
> > *How restrictive is the leader assumptions on the followers, i.e., cardinality, types, and utility, in practice? Do the authors see any path forward at relaxing these assumptions (cardinality and types)?*
>
> The assumption of a finite number of follower types with known utilities is less of a strong realism claim, and more of a standard modeling abstraction that matches how Stackelberg games are actually deployed in practice. For example, deployed security systems at LAX use Stackelberg games with a small catalog of (known) follower types (see [1]). From a theoretical standpoint, the assumption of finite follower types is also necessary in order to achieve no-regret guarantees in the worst case, due to existing lower bounds in the non-contextual Stackelberg game setting.
>
> With that being said, there is an interesting line of work on learning the follower’s utility/best-response function from repeated interactions in Stackelberg games (see, e.g. [2]), typically under the assumption that the follower’s type is fixed and does not change from round-to-round (unlike our setting). It may be possible to obtain positive results for learning the follower’s type in our contextual setting, under the assumption that the type can no longer change over time. Exploring this further is an interesting direction for future research.
>
> [1] Bayesian Stackelberg Games and Their Application for Security at Los Angeles International Airport by Jain et al. SIGecom exchanges, 2008.
>
> [2] Learning Adversary Behavior in Security Games: A PAC Model Perspective by Sinha et al., 2015
>
> > *Is it possible to say anything about the  approximation error in the regret due to the discretization?*
>
> Yes, it is possible to make the discretization error arbitrarily small by choosing delta to be arbitrarily close to zero in Definition 2.4. We let delta = O(1/T) so that the discretization error only adds an O(1) term to our regret bounds, but this term could be made to decay with T by picking, e.g. delta = O(1/T^2).

---

> > ### Comment · Reviewer_mHpn · 2025-11-20
> >
> > Thanks for the response. I maintain my positive score.

---

### Official Review · Reviewer_w4MJ · 2025-11-01

**Soundness:** 3
**Presentation:** 3
**Contribution:** 2
**Rating:** 6
**Confidence:** 3

**Summary:**

This paper treats the problem of online learning in Stackelberg games between a learner and a follower whose type changes over time. Both the leader and the follower have access to a $d$-dimensional side / contextual information variable $z_t$, and their payoffs are determined as follows:
- For the leader, as a function $u(z_t,a_t,b_t)$ of the context $z_t$ at time $t$ and the actions $a_t$ and $b_t$ of the leader and follower respectively.
- For the follower, as a time-varying function $u_t(z_t,a_t,b_t)$ of the context $z_t$ at time $t$ and the actions $a_t$ and $b_t$ of the leader and follower respectively. [The variability of $u_t$ over time is what the authors dub as the leader facing "a sequence of followers"]

The authors assume that the context $z_t$ or the follower's payoff function $u_t$ can be determined adversarially—but not both, as in that case, it is not possible to attain sublinear (Stackelberg) regret. In terms of information available to the players, they assume that the ensemble of the follower's payoff functions is known to the leader (though not the identity of each payoff function at each instance), the leader's payoff function could be either known or unknown to the follower, and the follower observes the leader's mixed strategy at each instance (so they can best-respond to it).

The authors provide a reduction to linear contextual bandits and, depending on whether the sequence of followers or contexts is adversarial, they provide the following guarantees:
1. *For adversarial contexts and stochastic followers:* an $\mathcal{O}(K\sqrt{T} \log T)$ regret bound when the recommendation subroutine in Algorithm 1 is the OFUL algorithm of Abbasi-Yadkori et al (2011).
2. *For adversarial follower sequences and stochastic contexts:* an $\mathcal{O}(K^2.5 \sqrt{T} \log T)$ when the regret-minimizing subroutine in Algorithm 1 is instantiated to the logdet-FTRL algorithm of Liu et al (2024).
The authors also provide a version of these theorems for the case where the leader's utility is unknown. In that case, the above bounds continue to hold in expectation with $K$ replaced by $d A_l A_f$, where $A_l$ is the number of actions available to the leader, and $A_f$ to the follower.

Finally, the authors describe how their results could be applied to the context of bidding in auctions with side information (Section 4.1) and Bayesian persuasion (Section 4.2).

Overall, I found the paper's topic to be on the fringe of ICLR topics—too specialized on a setting that is not central to generalist ML/AI audiences. My "weak accept" recommendation reflects my cautiously optimistic belief that the paper *could* be accepted if it garners sufficient support from experts in the domain. [I am knowledgeable in the general field but not an expert; still, I am skeptical regarding the difficulty of the reduction to contextual bandits or the novelty behind the meta-use of the .recommend subroutine in Algorithm 1.]

Otherwise, if the paper's contribution is not championed by an expert in the field, I believe this paper would be much better suited to a more specialized venue like EC. I will wait for the authors' rebuttal and the discussion with the rest of the review panel to converge.

**Strengths:**

The paper is well-written and, as far as I could tell, the analysis is sound and the results correct. I am not a dedicated expert in this literature but, as far as I can tell, the authors' contributions are positioned properly within the relevant literature.

**Weaknesses:**

My main objection to the paper is the paper's learning setting: I understand that the paper's framework may be sometimes referred to as "bandit learning" in the context of Stackelberg games but, otherwise, the assumptions made in the paper are fairly stringent and exacting. In particular, as was mentioned above, the authors assume that the ensemble of the follower's payoff functions is known to the leader, and the follower observes the leader's mixed strategy at each instance (so they can best-respond to it). Both assumptions go against the spirit of the partial information setting in bandits where it is assumed that players only observe their payoffs / losses—and even if this term is used in a part of the Stackelberg literature, it is a misnomer, which should not be propagated in a generalist venue like ICLR (or any of the other generalist learning venues, like NeurIPS, COLT, or ICML).

Other than that, as I said above, I found the paper's topic to be on the fringe of ICLR topics—too specialized on a setting that is not central to generalist ML/AI audiences.

If the paper's contributions are deemed sufficiently impactful by an expert in the field (I am not an expert on Stackelberg games and/or contextual bandits), I wouldn't find either of the above weakenesses as an obstacle to acceptance. Come what may however, I would insist on the authors' changing the "bandit learning" terminology to avoid clashing with established terminology in online / game-theoretic learning.

**Questions:**

None; see above.

---

> ### Author Response · Authors · 2025-11-19
>
> Thank you for taking the time to review our submission. Please find our responses to your questions and comments below.
>
> > *My main objection to the paper is the paper's learning setting: I understand that the paper's framework may be sometimes referred to as "bandit learning" in the context of Stackelberg games but, otherwise, the assumptions made in the paper are fairly stringent and exacting. In particular, as was mentioned above, the authors assume that the ensemble of the follower's payoff functions is known to the leader, and the follower observes the leader's mixed strategy at each instance (so they can best-respond to it). Both assumptions go against the spirit of the partial information setting in bandits where it is assumed that players only observe their payoffs / losses—and even if this term is used in a part of the Stackelberg literature, it is a misnomer, which should not be propagated in a generalist venue like ICLR (or any of the other generalist learning venues, like NeurIPS, COLT, or ICML).*
>
> Regarding your first point (ensemble of follower’s payoff functions is known to the leader): This can be seen as a way to impose structure on how the leader’s utility function may change from round-to-round in order to make the problem tractable. Indeed, structural assumptions on the form of the leader’s rewards are common (and usually necessary) in the literature on contextual bandits (see, e.g. [1], Chapter 7). In fact, the setting we study in Section 3.3 is a generalization of the well-studied linear contextual bandit problem.
>
> Regarding your second point (follower observes the leader’s mixed strategy), we are interested in designing no-regret strategies for the *leader* in our repeated Stackelberg game setting. Indeed, given the context and the leader’s mixed strategy, the follower’s best-response is well-defined, and so no learning is necessary on their behalf. As such, any references we make to “bandit learning” are with respect to the leader’s learning problem. This is the standard model in the literature on learning in Stackelberg games (e.g. 2-5 below). We are happy to make this point more explicit in the revision.
>
> [1]: Introduction to Multi-Armed Bandits by Aleksandrs Slivkins, 2019
>
> [2]: Vincent Conitzer and Tuomas Sandholm. Computing the optimal strategy to commit to. In Proceedings of the 7th ACM conference on Electronic commerce, pp. 82–90, 2006.
>
> [3]: Constantinos Daskalakis and Vasilis Syrgkanis. Learning in auctions: Regret is hard, envy is easy. In 2016 ieee 57th annual symposium on foundations of computer science (focs), pp. 219–228. IEEE, 2016.
>
> [4]: Maria-Florina Balcan, Avrim Blum, Nika Haghtalab, and Ariel D Procaccia. Commitment without regrets: Online learning in stackelberg security games. In Proceedings of the sixteenth ACM conference on economics and computation, pp. 61–78, 2015.
>
> [5]: Binghui Peng, Weiran Shen, Pingzhong Tang, and Song Zuo. Learning optimal strategies to commit to. In Proceedings of the AAAI Conference on Artificial Intelligence, volume 33, pp. 2149–2156, 2019.
>
> > *I found the paper's topic to be on the fringe of ICLR topics—too specialized on a setting that is not central to generalist ML/AI audiences*
>
> We respectfully disagree that our work is on the fringe of ICLR topics, and we would like to highlight some work on learning in games (including Stackelberg games) that has been published at ICLR recently:
>
> Generalized Principal-Agent Problem with a Learning Agent by Lin & Chen, ICLR 2025
>
> Feature-based online bilateral trade by Gaucher et al., ICLR 2025
>
> Approximating Nash Equilibria in Normal-Form Games via Stochastic Optimization by Gemp et al, ICLR 2024
>
> Learning Optimal Contracts: How to Exploit Small Action Spaces by Bacchiocchi et al., ICLR 2024
>
> Online Information Acquisition: Hiring Multiple Agents by Cacciamani et al., ICLR 2024
>
> Meta-learning in games by Harris et al., ICLR 2023
>
> Modeling content creator incentives on algorithm-curated platforms by Hron et al., ICLR 2023
>
> Global convergence of multi-agent policy gradient in markov potential games by Leonardos et al., ICLR 2022
>
> When Can We Learn General-Sum Markov Games with a Large Number of Players Sample-Efficiently? By Song et al., ICLR 2022
>
> Learning Graphon Mean Field Games and Approximate Nash Equilibria by Cui & Koeppl, ICLR 2022
>
> Linear last-iterate convergence in constrained saddle-point optimization by Wei et al., ICLR 2021
>
> Auction Learning as a Two-Player Game by Rahme et al., ICLR 2021

---

> > ### Comment · Reviewer_w4MJ · 2025-11-19
> >
> > Dear authors,
> >
> > Thank you very much for your response. Two quick replies from my end:
> > - Regarding terminology: As I said in my original review, I'm well aware that some papers dealing with Stackelberg games refer to the setting of your paper as "bandit learning". This doesn't make it right—or, more importantly, consistent with the situation it purports to model. "Bandit learning" means "learning with bandit feedback", and in your setting there is just too much information known at both ends (the leader and the follower) to justify this choice of terminology.
> > - Regarding the suitability of ICLR as a venue: I did not say that learning in games is a fringe topic for the leading ML/AI venues, I said that learning in *Stackelberg* games (the topic of your paper) is. The volume of the latter relative to the former is quite small, so I stand by my assessment that the right venue for this paper is EC or WINE—not ICLR. Going beyond game theory, if your paper treated—or could be extended to—a more general bilevel optimization framework, I might be inclined to concede this point, but not for the current setup.
> >
> > In view of all this, I remain on the fence regarding your paper, and will finalize my recommendation during the discussion with the rest of the reviewing panel.
> >
> > Thank you again for your replies. All the best,
> >
> > Reviewer w4MJ

---

> > > ### Author Response · Authors · 2025-11-19
> > >
> > > Thanks for your quick reply! We have two brief responses:
> > >
> > > We are happy to update our terminology if you have concrete suggestions. However we note that assuming the leader has knowledge of the set of possible follower types is very similar to the “transductive” setting in contextual bandits, where it is assumed that the learner knows the set of contexts it will encounter in advance. (E.g. Syrgkanis et al. (2016, a and b); Rakhlin & Sridharan (2016) in our submission.)
> > >
> > > Regarding the suitability of ICLR as a venue: We would like to highlight that our framework is applicable to the problem of learning to bid in auctions (Section 4.1), which is not a form of Stackelberg game.
> > >
> > > More broadly, we view our work and the paper which introduces the problem of learning in Stackelberg games with side information (Harris et al. 2024, published in NeurIPS) as belonging to a growing overlap between the economics & computation and machine learning communities.

---

### Author Response · Authors · 2025-12-02

Thanks for taking the time to review our work. We believe we've addressed most/all major criticisms in the rebuttal, and we believe our reply to Reviewer dzah has cleared up several misconceptions which may have led to their negative score.

---

### Meta-Review · Area_Chair_xyfR · 2026-01-06

**Summary:**

**1) Summary**
This paper studies online learning in contextual Stackelberg games where follower types vary over time and only bandit feedback is available. The core contribution is a reduction that maps the leader’s problem into linear contextual bandit learning in utility space, enabling near-optimal ( $\tilde O(\sqrt{T})$ ) regret in both adversarial-context/stochastic-follower and stochastic-context/adversarial-follower settings. The framework extends to unknown leader utilities and applications such as auctions and Bayesian persuasion.

**2) Strengths**

* Presents a clean and well-structured reduction from contextual Stackelberg games to linear contextual bandits, enabling off-the-shelf algorithms to achieve optimal regret up to logarithmic factors.
* Closes known gaps in regret bounds for bandit-feedback Stackelberg games, with extensions to unknown utilities showing robustness of the approach.
* Provides technically sound theorems with clear organization, supported by illustrative examples and applications to auctions and persuasion.

**3) Weaknesses**

* Assumptions are strong: the leader must know the full set of follower utility functions, and followers observe and best-respond to leader's mixed strategies, limiting realism and general applicability.
* The approach requires discretization and yields exponential per-round computational complexity, raising concerns about practicality even in small examples.

**Reviewer Concerns:**

It seems that most of the concerns have been addressed by the rebuttal, and since the ratings were originally more toward positive views, this paper would be projected to be accepted.

**Reviewer Scores:**

See the above.

---

### Decision · Program_Chairs · 2026-01-26

Accept (Poster)